

# The Furvela tent-trap Mk 1.1 for the collection of outdoor biting mosquitoes

Jacques D. Charlwood[1,2,3], Mark Rowland[1,3], Natacha Protopopoff[1,3] and Corey Le Clair[1,3]

[1] Department of Disease Control, London School of Hygiene & Tropical Medicine, University of London, London, United Kingdom
[2] Instituto de Higiene e Medicina Tropical, IHMT, Universidade Nova de Lisboa, UNL, Rua da Junqueira, Global Health and Tropical Medicine, GHTM, Lisbon, Portugal
[3] Pan African Malaria Vector Research Consortium (PAMVERC), Muleba, Tanzania

## ABSTRACT

Outdoor transmission of malaria and other vector borne diseases remains a problem. The WHO has recently recognized the need for suitable methods for assessing vector density outdoors and a number of tent-traps have been developed. Only one such trap, the Furvela tent-trap, does not require an 'entry' behavior on the part of the mosquito. It remains the cheapest and lightest tent-trap described. It takes less than two minutes to install and is the only trap that uses readily available components. Here we describe recent modifications to the trap, which make it even easier to set up in the field, provide a standard operating procedure (SOP) and describe some recent experiments examining the effect of the addition of light and door placement to working of the trap. The trap provides the closest approximation to CDC light-traps, widely used to collect indoor biting mosquitoes. This enables the effect of both indoor and outdoor interventions on mosquito density and behavior to be determined.

## INTRODUCTION

With the current drive to eliminate malaria, worldwide reductions in the disease have occurred (*Bhatt et al., 2015*), and according to the WHO the incidence of the disease, which takes into account population growth, is estimated to have decreased by 37% between 2000 and 2015 (*WHO, 2016*).

This success has highlighted the challenges that remain, in particular the control and elimination of residual outdoor transmission not controlled by long lasting insecticide treated nets (LLIN) or Indoor Residual Spraying (IRS) with insecticide. Whilst the monitoring of indoor transmission remains important, outdoor transmission needs to be assessed. Indeed, at the 70th World Health Assembly in May 2017, WHO Member States expressed strong support for the strategic approach proposed in the Global Vector Response 2017–2030 which states that: 'Assessments of vector populations should use up-to-date methods and techniques to ensure that results are informative for guiding and assessing vector control. Of particular need are robust indicators for vector-borne disease

Corresponding author
Jacques D. Charlwood,
jdcharlwood@gmail.com

risk, especially in low transmission settings, and methods for assessing vector behaviour such as mosquito outdoor biting.' (*WHO, 2017*). Although it is malaria vectors that are of primary concern, many other outdoor biting (exophagic) mosquitoes are potential vectors of pathogens, including a number of 'emerging' diseases.

The objective of monitoring outdoor biting would be to assess vector species composition, abundance, and the time and place of biting. The primary requisites of such methods are that they catch mosquitoes which would normally bite people outside, using simple and inexpensive equipment. Outdoor exposure, at least in the evening, is best measured in human landing collections (HLC), in which mosquitoes are caught attempting to bite the exposed lower legs of collectors sitting outside, as this is usual behaviour. HLC have been used extensively in the past, largely to sample malaria vectors (*Silver, 2008*). Landing collections, require considerable supervision however, and, since mosquitoes may be able to inject pathogens before being caught, impose risks to the collectors. Mosquitoes are also attracted to different humans at different rates. Indeed, this differential attraction may also differ by species (*Knols et al., 1995*).

In a number of studies commercially available Centre for Disease Control (CDC) light-traps have been used outdoors because they do not expose people to mosquito bites, are widely available and cheap to run (*Githeko et al., 1994*; *Costantini et al., 1998*; *Cooke et al., 2015*). Whilst operationally practical, evidence from these studies suggests that they do not adequately sample the outdoor biting fraction of malaria vectors (*Fornadel, Norris & Norris, 2010*). Thus, *Costantini et al. (1998)* reported no significant correlation between *Anopheles gambiae* s.l., HLC outdoors and outdoor CDC light-traps, and density-dependent correlation in the case of *Anopheles funestus. Cooke et al. (2015)* attempted to measure the outdoor biting fraction of the population by employing a CDC light-trap hung adjacent to an occupied, open-sided rain shelter constructed from a domed one-man tent, but concluded that such traps were a 'limitation' of their study. Other traps, notably the MM-X trap (*Njiru et al., 2006*) and more recently the 'Suna' trap (*Homan et al., 2016*) have been developed to catch outdoor biting mosquitoes. Nevertheless, they do not provide an easily quantifiable estimate of exposure, nor are they cheap or easily available. Tent-traps are the simplest alternative solution.

*De Meillon (1934)* first used tent-traps to collect *An. gambiae* s.l. in South Africa. He used a plastic gazebo that had openings cut close to the roof that mimicked the eaves in a house. Collectors stood inside the gazebo (their breath and odour attracting mosquitoes) and collected them from the inside walls when the insects were inter-current resting (*Mattingly, 1965*). Thus, this was more akin to a moveable experimental hut than a trap for collecting outdoor biting mosquitoes.

More recently a number of other tent-traps have been developed (*Charlwood, 2005*; *Govella et al., 2009*; *Krajacich et al., 2014*). All but one, however, are similar to the gazebo of DeMeillon in that they require an 'entry' behaviour on the part of the mosquitoes for them to be caught. The usefulness of these tent-traps is limited because not all mosquitoes go inside houses and, as pointed out by *Gillies (1974)* 'the effectiveness of a baited trap for a particular species of fly primarily depends on its responses to the trapping device in the presence of the attractant stimuli used'.

The 'Furvela' tent-trap catches mosquitoes before they enter the tent, and is therefore likely to sample the 'true' outdoor fraction of the population. The trap remains the most straightforward, lightest and cheapest tent-trap available. It is the only trap that is made from commercially available 'off the shelf' components, including the tent itself. Collections in the Furvela tent-trap are closely correlated to CDC light-trap collections used to monitor indoor biting mosquitoes (*Govella et al., 2009*; *Charlwood et al., 2011*; *Charlwood et al., 2012*), this makes it especially useful for the measurement of changes in indoor/outdoor ratios following the application of methods to control indoor biting mosquitoes. Since changing the collection bag is easy with the Furvela tent-trap it is also possible to examine the biting profile of the mosquitoes throughout the night as was done previously in Ghana (*Charlwood et al., 2011*).

The trap has been used to make the first map of the spatial variation in outdoor biting densities of mosquitoes (*Charlwood et al., 2013*) and has recently been used to evaluate an intervention that targeted outdoor biting mosquitoes in Cambodia (*Charlwood et al., 2016*). Presently they are being used to monitor outdoor biting mosquitoes over a 950 km$^2$ area in 40 villages in Kagera Province, Tanzania.

Since its initial description (*Charlwood, 2005*; *Govella et al., 2009*) the trap has undergone a number of modifications which make it even easier to set up in the field, but do not affect its basic operation. Here we describe these modifications and discuss some recent experiments, including an examination of the effect that the addition of a light to the trap and the effect that door position relative to the sleepers' head has on numbers caught. We supply evidence that the trap does indeed catch outdoor biting mosquitoes and provide a Standard Operating Procedure (SOP) on how to set the trap up in the field.

## METHODS

The basic principle of the Furvela trap is that host odour and exhaled gases emanating from a gap, the diameter of a CDC trap, in the predominantly closed door of the tent are sucked into a CDC trap (without the light, lid or grid) placed, outside the tent, horizontally between 2 to 3 cms from the opening in the door. On approach to the opening the insects are sucked into the trap and held in the standard CDC trap conical collection bag. The suction from the fan effectively prevents any mosquitoes from entering the tent, even at very high densities, so that the sleeper is only exposed if they leave it for some reason. As originally described (*Charlwood, 2005*; *Govella et al., 2009*) the setting up of the trap was slightly awkward. Recent improvements to the original trap include the following (Figs. 1A–1F):

A. *The opening is more easily standardised.* For this the sides of the tent are sewn back rather than being folded back by clips (although clips can still serve). In addition to standardising the opening, sewing it makes it more difficult for the tent to be zipped up.

B. *Attaching the trap is easier.* A small hole is made, using hot wire, in the Perspex close to the top of the CDC trap. Short (6–7 cm) lengths of the same wire are threaded through these holes and medium sized folding-clips attached. The clips are used to attach the trap to the tent.

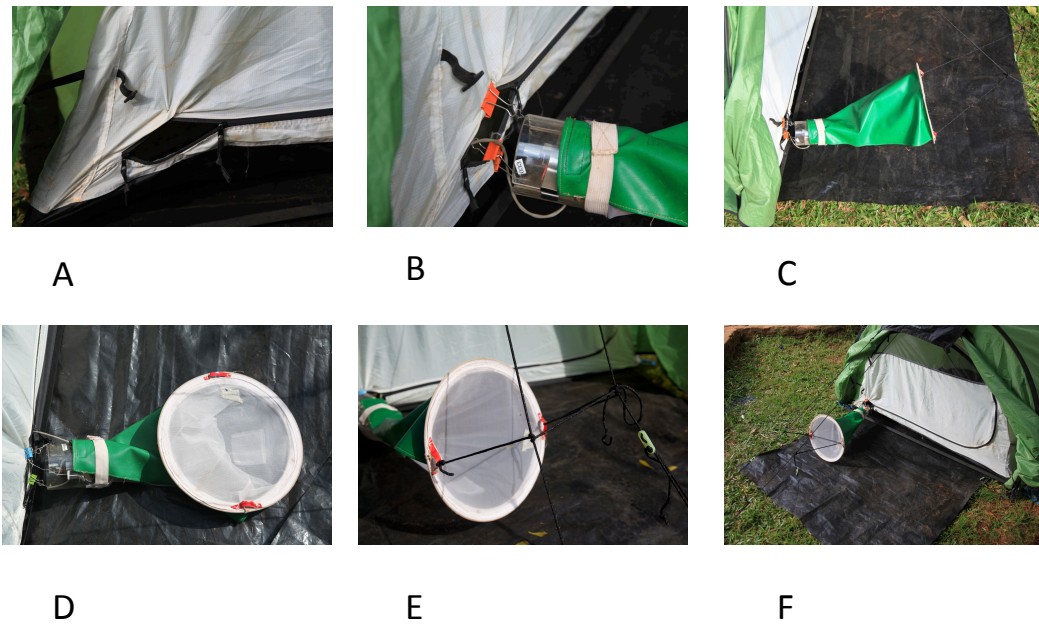

**Figure 1 The Furvela tent-trap Mk 1.** Modifications and installation of the trap—(A) the opening of the tent door is sewn open; (B) clips are used to attach the body of the trap to the tent; (C) a rain-proof cover is added to the collection bag; (D) eyelets are sewn into the back of the collection bag; (E) the collection bag is suspended using guy ropes attached to the tent; (F) a footprint that extends under the trap is added.

C. *A cover over the collection bag allows collections to continue in the rain.* The collection bag has a rain cover (that attaches to the body of the trap with Velcro or a rubber band) sewn over the top half of the net.

D. *Supporting the collection bag is easier.* Two eyelets are sewn into the bottom of the bag to facilitate attachment.

E. *An external support for the collection bag is no longer required.* The bag is now supported in place by two guy ropes, thus eliminating the need for an external support for the bag. Some tents (e.g., the Hoolie Wildcountry) already have these guy ropes available. They generally are otherwise not difficult to attach to other tents.

F. *A footprint facilitates usage.* Although not mandatory, we find that a plastic sheet ('footprint') under the tent and up to the edge of the collection bag prevents holes in the bottom of the tent and may reduce exposure of the collected mosquitoes to ants. The Standard Operating Procedures (SOP) (Supplemental Information 1) shows these modifications in more detail. The installation of the trap can be seen in the video: https://www.youtube.com/watch?v=irgBPrDQ2Pw. When not in use as a tent-trap the CDC trap can easily be reconverted to a standard light-trap without the need to remove the clips (which hang outside of the trap body).

A series of experiments were undertaken to determine the relationship between different indoor trapping types with the tent-trap and to investigate possible procedures that might increase the efficiency of the trap. The experiments took place in Kyamyorwa village, located on an inlet of Lake Victoria, in Kagera Province, northern Tanzania before and

after an intervention to control malaria was undertaken in the village. The intervention included indoor residual spraying of insecticide for the control of post-prandial insects and the introduction of long-lasting insecticidal nets for the control of biting insects. Indoor walls of houses in the village were sprayed with pirimiphos-methyl (Actellic) and an LLIN incorporating permethrin and the synergist piperonyl butoxide (PBO) were distributed to residents in February 2015.

In addition to CDC light-traps, exit window-traps are another possible proxy for exposure indoors. The number of mosquitoes collected from a $50 \times 50 \times 50$ cm, netting sided, window-trap (*Silver, 2008*) that covered the only window of a bedroom occupied by two adults was compared to the number collected in a tent-trap 20 m from the house occupied by a single sleeper. The tent-trap was operated twice a week and window-trap collections were undertaken on a daily basis over a nine-week period in Kyamyorwa. Mean numbers per trapping method per ISO week were compared using Pearson's correlation in Excel (Supplemental Information 2).

In order to feed inside a house mosquitoes need to enter through relatively small openings (such as the gap between the eaves and the roof). Not all species of mosquito will do this and so these species are mainly caught biting outdoors. One way of determining if the Furvela tent-trap catches outdoor biting mosquitoes is to determine if these mosquitoes are caught in the trap. We, therefore, compared the species ratios of mosquitoes collected indoors with CDC light-traps or window-traps with outdoor collections in the tent-trap in different situations. In order to determine if the numbers collected varied in the same manner during the night, mosquitoes were removed from the window trap at four-hour intervals (22:00, 02:00 and 06:00), whilst at the same time the collection bag on the tent-trap was changed. Collected anophelines were identified (by JDC) using the keys of *Gillies & De Mellion (1968)* and *Gillies & Coetzee (1987)*. Non-anophelines from Mozambique were kindly identified by Dr Ralph Harbach of the British Museum of Natural History, London, those from Tanzania were identified by the authors using the descriptions provided by *Gillett (1972)*.

As part of the Pan African Malaria Vector Research (PAMVERC) trial (*Protopopoff et al., in press*) both tent-traps, outdoors, and light-traps, indoors, are being used to sample malaria vectors in the 48 clusters of the trial area. During each round of sampling, the traps are set up for one night in seven randomly selected houses per cluster. Houses for light-trap and tent-trap samples were chosen at random from the census database obtained at the start of the study. During sampling the light-trap was set up in a bedroom at the end of a bed in which someone slept under a mosquito net and, by one house, a tent-trap was set up (and slept in by the PAMVERC entomologist/collector). Collections from the baseline year (2015) were analysed and are presented. Data were entered into a database and analysed with Stata 12 (*Stata, 2013*). Since the data were over-dispersed (the deviance was greater than the mean), differences in mosquito density between the two collection methods (light trap or tent trap) were estimated using negative binomial regression. Standard errors were adjusted to allow for within-cluster correlation of responses using robust standard errors.

Other factors, including number of sleepers, chemical lures, light or the position of the sleepers' head relative to the opening, may affect the efficiency of the trap. Two of these

variables were investigated in Kyamyorwa village: the effect of a light source and the effect of door position on the number of mosquitoes captured. The effect of a light source was investigated in a series of collections using four tent-traps in Kyamyorwa (Supplemental Information 3).

A new moon occurred on June 27 2014 (the start of the experiment) and there was little/no ambient illumination during collection dates, providing optimal experimental conditions. We used 2-door tents for these experiments (the Highlander Glen Orchy 2 Tent®). A standard tent-trap functioned as a control on one side of the tent while the trap on the other door incorporated an incandescent bulb, as used in the CDC light-trap. The trap with the light was rotated between sides on alternate days. Tent-traps were operated from 21:00–06:30 the following day and were operated from June 30–July 4, 2014. The incidence rate ratio (IRR) and density rate ratios (DRR) (i.e., a relative difference measurements used to compare the incidence rates of events occurring at any given point in time) were used to compare the relative density of mosquitoes sampled by a tent-trap with light (TT + L) with a standard trap (TT). Variables including collector, collection date, and sampling site were identified as potential confounding factors during univariate analysis and were included in the final regression model.

Since the trap relies on the breath and odour of the host inside the tent, the relative position of the trap to the hosts head might influence the number of mosquitoes collected. We, therefore, determined if the position of the door (at the side or the front) affected numbers collected. The tents used were the two-door Glenn Orchy Highlander (with doors at the sides) and the Taurus Ultra-light two-man tent (with a single door at the front of the tent). Collectors rotated between tents on alternate nights and the tents were rotated every second day.

## ETHICS

The collections conducted in Tanzania were done as a component of the Pan African Malaria Vector Research Consortium project 'Evaluation of a novel long lasting insecticidal net and indoor residual spray product, separately and together, against malaria transmitted by pyrethroid resistant mosquitoes' which received ethical clearance from the ethics review committees of the Kilimanjaro Christian Medical College (certificate number 781 on the 16 September 2014), the Tanzanian National Institute for Medical Research (20 August 2014), and the London School of Hygiene and Tropical Medicine (reference 6551 on 24 July 2014). The trial was registered with ClinicalTrials.gov (registration number NCT02288637) on 11 July 2014.

Collections from Mozambique were undertaken under the aegis of the joint Instituto Nacional de Saúde (INS)–DBL Centre for Health Research and Development project 'Turning houses into traps for mosquitoes', which obtained ethical clearance from the National Bioethics Committee of Mozambique on 2 April 2001 (ref: 056/CNBS/01).

Prior to beginning collections, informal sensitisation sessions were conducted with village members to explain sampling-related activities. Written and oral informed consent was obtained from all participants who could withdraw from the study at any time should they wish to do so.
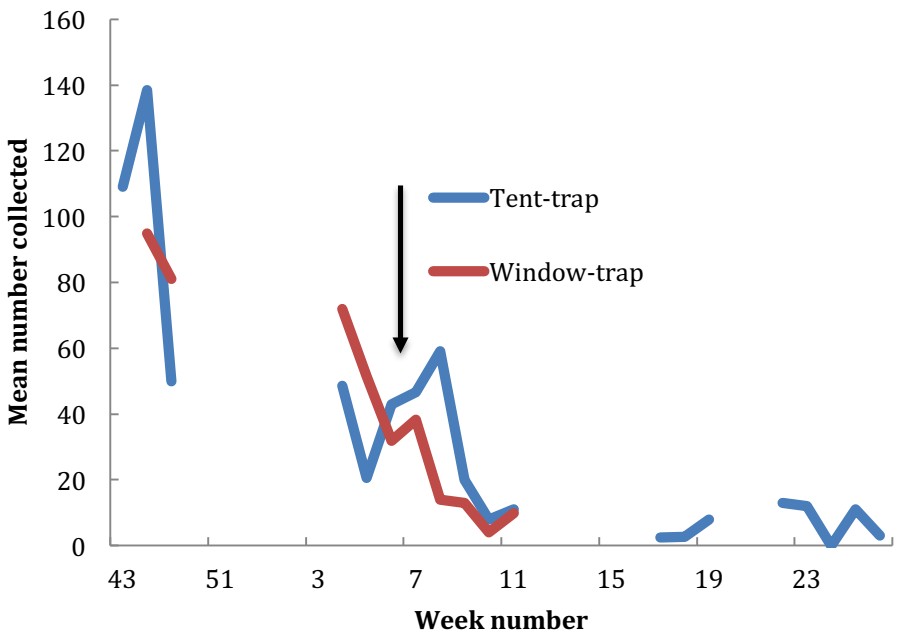

**Figure 2** **Exit window trap and Furvela tent-trap collections of *An. gambiae* s.l. from the village of Kyamyorwa, Muleba District, Kagera Region, Tanznia.** The arrow marks the time when the interior walls of the bedroom were sprayed with pirimiphos-methyl (Actellic) at 1 g ai per m$^2$ (prior to the spray cross-correlation between mean weekly numbers in the window trap and numbers in the tent-trap $r = 0.93$, $p \geq 0.001$).

## RESULTS

### Indoor outdoor/ratios: window traps versus Furvela tent-trap

In the baseline year, between 23 October 2014 (ISO week number 43) and 14 March 2015 (ISO week number 25), 70 collections were undertaken from the window trap (only two collections being undertaken in December) and 17 tent trap collections were performed. Changes in population density obtained from the two collection methods were similar ($r = 0.93$, $P < 0.001$) (Fig. 2). Numbers in the window trap declined from a peak of 1,018 on the 24 November 2014 to single figures in the second week of February and from 243 in the tent-trap (on the 19 November 2014 to 20 on the 25 February 2015). The decline in both collections fitted a logarithmic series (window $y = -34.45\ln(x) + 120.82$, $R^2 = 0.793$; tent $y = -37.93\ln(x) + 126.75$, $R^2 = 0.806$). The house was sprayed with pirimiphos-methyl on 23 February 2015 and subsequently the number in the window-trap fell to zero but numbers in the tent-trap persisted, albeit at a very low density. A larger proportion of the catch was caught in the earlier part of the night in the tent-trap compared to the window-trap ($X^2 = 16.8$, $df = 3$, $P = 0.014$) although, for both collections, most insects were caught in the middle hours of the night.

*Anopheles gambiae* was caught in approximately equal numbers from window-trap and tent-trap whilst other species including *Coquelletidia fuscopennata*, *Mansonia* spp. and *Culex* spp. were collected in greater numbers in the tent-trap compared to the window-trap (Fig. 3). Comparable results, between CDC light-trap and Furvela tent-traps, were obtained

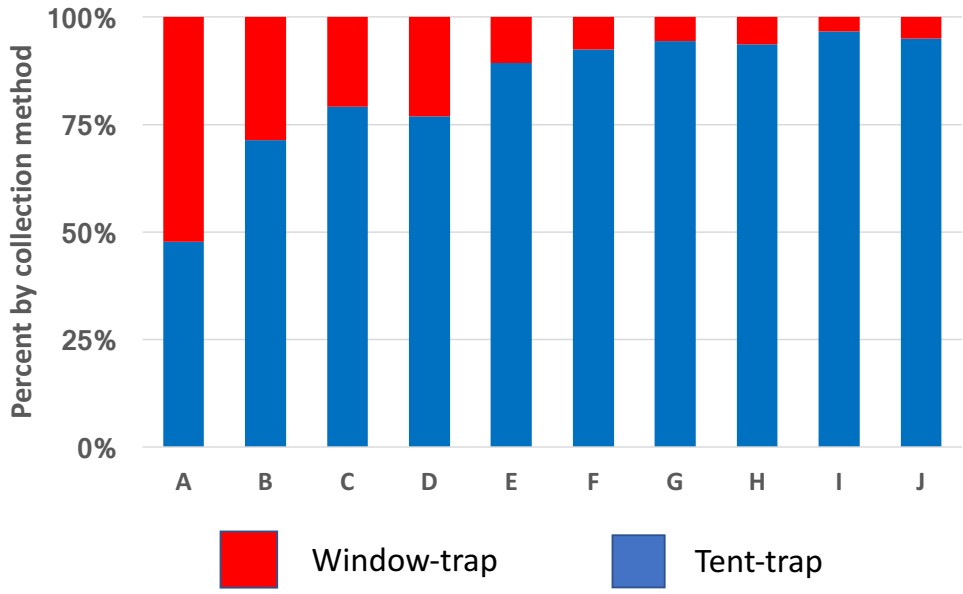

**Figure 3** **Species comparison between window-traps and Furvela tent-traps.** Window-trap tent-trap ratios of mosquitoes from Kyamyorwa, Muleba District, Tanzania. (A) *Anopheles gambiae* ($n = 10,512$), (B) *An. funestus* ($n = 81$), (C) *An. coustani* ($n = 27$), (D) *An. zeimanni* ($n = 282$), (E) *Cx. quinquefasciatus* ($n = 471$), (F) *Cx. tritaeniorhynchus* ($n = 21$), (G) *Coquelettidia fuscopennata* ($n = 130$), (H) *Mansonia spp.* ($n = 737$), (I) *An. squamosus* ($n = 81$) and (J) *An. pharoensis* ($n = 49$).

**Table 1** **Tent-trap and light-trap comparisons from the PAMVERC trial in Muleba District, Tanzania.**

|  | CDC light-trap | | | Furvela tent-trap | | | Density ratio | 95% CI | P value |
|---|---|---|---|---|---|---|---|---|---|
|  | Number of collections | Mean | 95% CI | Number of collections | Mean | 95% CI | | | |
| All vectors | 3,395 | 1.9 | 1.4–2.5 | 495 | 2 | 1.3–2.6 | 1 | 0.8–1.2 | 0.888 |
| All mosquitoes | 3,364 | 8.4 | 5.8–11.0 | 491 | 11.4 | 7.8–15.0 | 1.4 | 1.2–1.5 | <0.001 |
| *An. gambiae* s.l. | 3,395 | 1.7 | 1.2–2.2 | 495 | 1.8 | 1.2–2.5 | 1.1 | 0.9–1.4 | 0.356 |
| *An. funestus* | 3,395 | 0.3 | 0.2–0.4 | 495 | 0.14 | 0.07–0.2 | 0.5 | 0.3–0.7 | <0.001 |
| *An. zeimanni* | 3,395 | 0.2 | 0.1–0.3 | 493 | 0.37 | 0.2–0.6 | 2.1 | 1.0–4.4 | 0.058 |
| *Cx. quinquefasciatus* | 3,394 | 3.1 | 1.4–4.8 | 494 | 3.7 | 1.8–5.5 | 1.2 | 0.9–1.5 | 0.197 |
| *Mansonia* sp. | 3,390 | 1 | 0.7–1.3 | 495 | 1.5 | 0.9–2.2 | 1.6 | 1.3–1.9 | <0.001 |
| *Cq. fuscopennata* | 3,391 | 1.5 | 1.1–1.8 | 495 | 2.6 | 1.8–3.4 | 1.8 | 1.3–2.3 | <0.001 |

during the first year of the PAMVERC trial when 34,092 mosquitoes were collected from 3,395 light-trap collections and 495 tent-traps (Table 1).

A similar implication comes from the redrawn data from Massavasse in Mozambique, where 144,317 mosquitoes were collected from 2,551 light-trap and 94,354 from 776 tent-trap collections (Fig. 4) (*Charlwood et al., 2013*). In Massavasse only *An. funestus* and *Culex* spp. (mostly *Cx. quinquefasciatus*) were caught in greater numbers indoors compared to the other species shown in the figure. Following the application of the

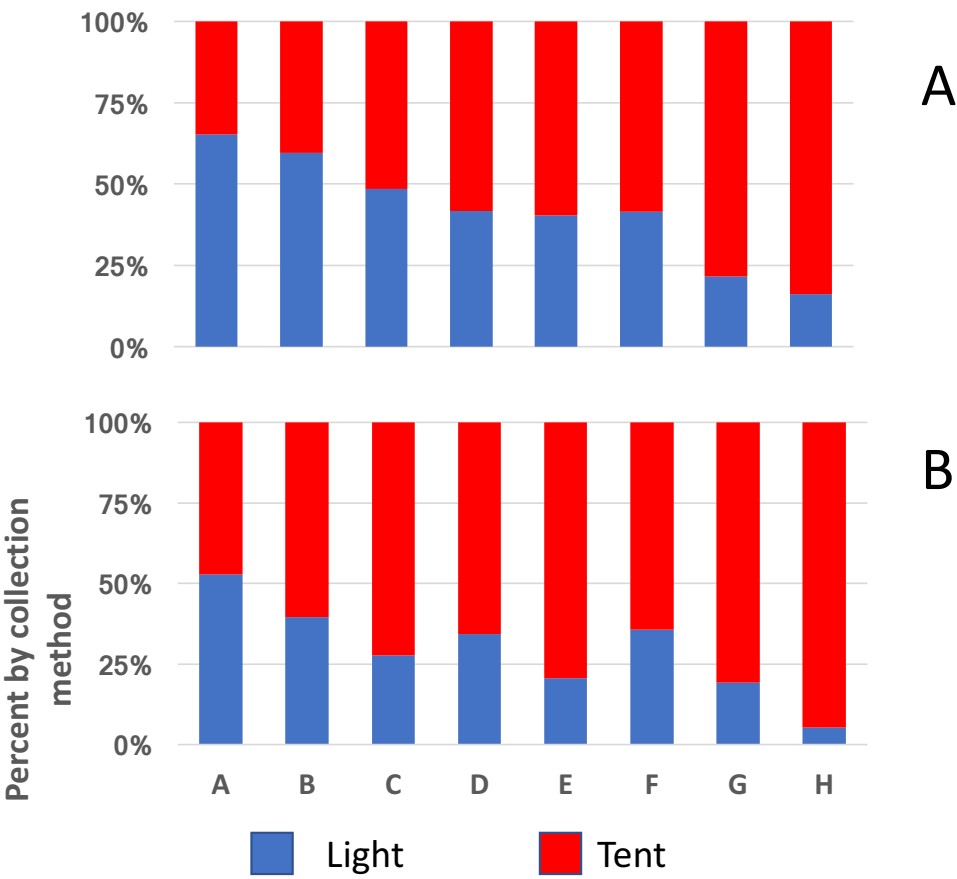

**Figure 4 Indoor and outdoor ratios of mosquitoes from Massavasse.** Indoor and outdoor ratios of the principal mosquitoes collected in Massavasse, Chockwe District, Gaza Province, Mozambique, before (A) and after the walls were sprayed with bendiocarb at 0.4 gm ai m$^2$ (B). (A) *Anopheles funestus* (*n* before spray = 3,754, *n* after spray = 1,742); (B) *Culex* spp.—mainly *Cx. quinquefasciatus* (*n* before = 4,267, *n* after spray = 2,208); (C) *An. pharoensi* s (*n* before = 2,642 *n* after spray = 2,078); (D) *An. arabiensis* (*n* before = 5,406, *n* after spray = 512); (E) *Aedes scatopagoides* (*n* before = 1,085 *n* after spray = 395); (F) *Mansonia africana* (*n* before = 38,173, *n* after spray = 12,054). Data from *Charlwood et al. (2013)*.

insecticide bendiocarb to the interior walls of houses in the village a greater proportion of the collection of all species, including the exophagic ones, was obtained in tent-traps (Fig. 4).

### Effect of light on numbers collected

Thirty-two collections (standard tent-trap (TT) *n* = 16, and tent-trap+light (TT + L) *n* = 16) were performed. A total of 180 *An. gambiae*, 104 *Mansonia* spp., 195 *Cq. fuscopennata* and, 140 *Culex* spp. were collected over a 4-day period. Data fit a negative binomial distribution. Surprisingly, the TT+L caught significantly fewer *Anopheles* females than the TT (Adjusted-IRR = 0.56, *P* < 0.001) (Table 2).

### Effects of door position on numbers collected

The rate ratio of the total number of *Anopheles* captured (IRR 1.05, 95% CI [0.52–2.12], *P* = 0.9) after 16 collections was not significantly different between the tents, when

**Table 2  Comparison between number of mosquitoes collected between tent-trap and tent-trap with a light source.** Comparison between the density of species collected in a standard tent-trap (TT) and tent-trap with an incandescent 6 V light, as used in a standard CDC light-trap, (TT + Light), Kyamyorwa, Muleba, Tanzania IRR = Incidence Rate Ratio; Adjusted = IRR adjusted for collector, location, and date; 95% CI = 95% Confidence Interval.

| Method | Anopheles* | | Mansonia sp | | Total all species | |
|---|---|---|---|---|---|---|
| | DRR | Adjusted DRR* | DRR | Adjusted DRR* | DRR | Adjusted IRR* |
| Tent-trap | 1.0 | 1.0 | 1.0 | 1.0 | 1.0 | 1.0 |
| Tent-trap + Light | 0.24 | 0.14 | 0.56 | 0.52 | 0.47 | 0.38 |
| 95% CI | 0.08, 0.57 | 0.07, 0.28 | 0.2, 1.6 | 0.27, 0.97 | 0.22, 1.02 | 0.24, 0.59 |
| p-value | 0.002 | <0.001 | 0.282 | 0.044 | 0.056 | <0.001 |

differences in host attractiveness were taken into account. The IRR of the total number of all mosquitoes captured (IRR 0.92, (95% CI [0.50–1.68]), $P = 0.77$) was also not significantly different between the tents, when differences in host attractiveness were accounted for. Thus, door position does not affect the efficiency of the trap (Supplemental Information 3).

## DISCUSSION

In order to determine the density of potential vectors biting outdoors suitable alternative methods to 'gold standard' HLC are required. This is because, in addition to being expensive and requiring considerable supervision, mosquitoes may transmit pathogens before being caught by collectors performing HLC and so are at risk of becoming ill. The Furvela tent-trap is one such alternative.

Like all trapping techniques that rely on the attraction of mosquitoes to a human, including HLC and CDC light-traps, it is likely to be affected by the individual attractiveness of the humans acting as bait. Nevertheless, the greater diversity of species collected in Furvela tent-traps, from a number of study sites, compared to the diversity from indoor collections, especially CDC light-traps, indicates that it adequately samples outdoor biting mosquitoes (*Govella et al., 2009*; *Charlwood et al., 2012*). A pairing of either a CDC light-trap or window-trap (used to collect endophagic mosquitoes) with a Furvela tent-trap (to collect exophagic ones) enables the effect of environmental perturbations or interventions to be determined.

The data obtained to date with the tent-trap confirm that arbovirus vectors, like *Cq. fuscopennata* or *Mansonia* spp., bite predominantly, but not exclusively, outdoors. Among malaria vectors the members of the *An. gambiae* complex tend to be collected in equal numbers in tent and light-traps whilst *An. funestus* have generally been caught in light-traps at higher rates compared to tent-traps, confirming their endophilic status. Whether the apparent change from endophilic and endophagic, behaviour to exophagic behaviour in *An. funestus* following IRS with bendiocarb in Massavasse was because the insects entering houses were killed before being caught in light-traps, or because they refrained from entering in the first place, remains unknown and merits further investigation.

Mosquito populations, as assessed by indoor collections with CDC light-traps, are often considered to be temporally unpredictable. This may be due to environmental factors,

such as rainfall, that may affect the proportion of the population biting indoors (and so available for capture by the light-trap). For example, in Ghana a greater proportion of the night's catch of *An. coluzzii* were collected indoors on rainy nights (*Charlwood et al., 2011*). The total collected in paired indoor light-trap and outdoor tent-trap collections was, however, not different from the number expected. Subsequently the collections returned to the anticipated ratios. In order to control for potential changes in the proportion of the mosquito population biting indoors or outdoors, studies assessing mosquito population dynamics should include simultaneous indoor and outdoor collections. Paired collections may reduce some of the 'noise' in such data making it more amenable to analysis.

Vector control plays a big part in current efforts to eliminate malaria. As a result, mosquito densities may become very low. Improving capture efficiency may be useful in such situations. The addition of a light to the trap, however, actually reduced the numbers. This would suggest that the light actually had a repellent effect on the local mosquitoes but why this should be so is not known. Chemical lures may enhance collections and merit consideration.

The Furvela tent-trap weighs as little as 2.5 kg. Since both CDC light-traps and Furvela tent-traps are portable, effective surveillance, using a limited number of traps of both indoor and outdoor biting mosquitoes over considerable areas is possible.

Mosquito populations can vary as much in space as they do in time (*Magbity & Lines, 2002*). Determination of high density areas (so called 'hot spots') may enable focussed control, such as targeting selected water bodies for larval control, to be undertaken. In order to determine where high density areas occur maps of mosquito density are required. Ease of transportation makes the Furvela tent-trap particularly suitable for mapping studies. Mapping using tent-traps enables locations to be determined according to geographical co-ordinates rather than being dependent on where appropriate houses are available for the installation of light-traps. Among mosquitoes that have fixed breeding sites, such as *An. funestus*, such information may allow estimates of flight range to be obtained, which may also help determine how wide a potential *cordon sanitare* needs to be for it to be successful (*Charlwood et al., 1998*).

A CDC-trap costs 120 US$ and the cone collection bag 18 US$ at current prices. A 6V 4.5Ah lead acid rechargeable battery, that costs circa 15 US$ and weighs 0.7 kg, can power the trap for a night (see the SOP). Two man tents can cost less than 25$. Since the location of the door does not affect collections the choice of tents is large. Simple tents weigh less than 1.8 kilos, hence the total weight of the trap (including the tent) is just over 2 kg and costs 173 US$. The professionals who might want to monitor mosquitoes are likely to have CDC light-traps and batteries available. In this case, the trap would cost just 43 US$. It only requires one person to put it up and, because the interior of the tent is not altered in any way, is comfortable for the sleeper. As can be seen in the video, once the tent itself is up it takes just a few minutes to install. The ease with which collection bags can be changed means that collections can easily be sub-divided throughout the night. As a routine, they can be changed when local residents enter their houses, so that estimates of actual outdoor

exposure can be obtained. When not in use the CDC trap can easily be reconverted to a standard light-trap without the need to remove the clips, which hang outside of the trap (see the SOP).

Passive monitoring of mosquito populations is providing information on the distribution of mosquitoes in Europe (*Kampen et al., 2015*). Presently this is restricted to the collection of insects indoors. The use of the Furvela tent-trap need not be confined to the tropics or to professionals. The very simplicity of the trap means that anyone who goes camping can collect, without risk to themselves, from local outdoor biting fauna. Using a smartphone, collections can be geo-referenced and the locality photographed. Thus, with a minimum amount of professional resources and data collection, national databases (of such things as bird flu vectors) could be established. Data, and eventually samples, might be sent to a central location (such as a Mosquito Abatement Office) where they would be identified and processed. Unlike all other tent-traps the extra equipment required by any sleeper is minimal.

## CONCLUSIONS

Monitoring outdoor biting activity of malaria vectors is an important component of present efforts attempting to control the disease. Our understanding of the ecology of mosquitoes which may be vectors of emerging diseases, other than malaria, is limited. These mosquitoes may well be exophagic. The WHO has recently recognized the necessity for novel sampling tools to conduct surveillance of outdoor biting mosquitoes with the objective of assessing vector species composition, time and place of biting, and abundance (*WHO, 2014*). Furvela tent-traps are a simple and effective way of collecting such mosquitoes.

## ACKNOWLEDGEMENTS

We would like to thank everyone who has slept in the tents during the studies described here. Special thanks to Elsa Tomás for her help in the design of the original trap and for her ongoing contribution to collections in it. Thanks too, to Judith Cronery for the suggestion to sew open the tent opening. Previously unpublished data form part of the MRC supported PAMVERC project in Muleba, Tanzania. We would also like to acknowledge the improvements to the original manuscript due to the referees and Dominique Shoham for copy editing the manuscript.

### Funding
This work was supported by the Joint Global Health Trials Scheme by DFID, Medical Research Council and Wellcome Trust (Grant Ref: MR/L004437/). The funders had no role in study design, data collection and analysis, decision to publish, or preparation of the manuscript.

## Grant Disclosures

The following grant information was disclosed by the authors:
Joint Global Health Trials Scheme by DFID.
Medical Research Council and Wellcome Trust: Ref: MR/L004437/.

## Competing Interests

The authors declare there are no competing interests.

## Author Contributions

- Jacques D. Charlwood conceived and designed the experiments, performed the experiments, analyzed the data, wrote the paper, prepared figures and/or tables, reviewed drafts of the paper.
- Mark Rowland wrote the paper, reviewed drafts of the paper.
- Natacha Protopopoff performed the experiments, reviewed drafts of the paper.
- Corey Le Clair conceived and designed the experiments, performed the experiments, analyzed the data, wrote the paper, reviewed drafts of the paper.

## Human Ethics

The following information was supplied relating to ethical approvals (i.e., approving body and any reference numbers):

This study was approved by the ethics review committees of the Kilimanjaro Christian Medical College (certificate number 781 on 16/09/2014), the Tanzanian National Institute for Medical Research (20/08/2014), and the London School of Hygiene and Tropical Medicine (reference 6551 on 24/07/2014). The trial is registered with ClinicalTrials.gov (registration number NCT02288637) on 11/7/2014. Prior to beginning collections, informal sensitization sessions were conducted with village members to explain sampling-related activities. Written informed consent was obtained from all participants who could withdraw from the study at any time should they wish to do so.

## Data Availability

The raw data has been supplied as Supplemental Files.

## Supplemental Information

Supplemental information for this article can be found online at http://dx.doi.org/10.7717/peerj.3848#supplemental-information.

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
