# Peer review of "The Furvela tent-trap Mk 1.1 for the collection of outdoor biting mosquitoes"

_PeerJ, doi:10.7717/peerj.3848_

## Round 0.1 · original submission · Major Revisions

# Staff note - the article has not been sent for Peer Review yet, therefore some of the templated text in this email is incorrect #

Dear Dr Charlwood

Your article is interesting and is likely to constitute a valuable contribution to the field. However, I believe some revision is necessary before your article can be sent out for peer-review.

As it stands, it is quite confusing to understand what your manuscript entails. The aims are not clearly stated in the abstract or in the Introduction.

In the text, first you provide a list of improvements, but it is not clear whether you were the first to make these (i.e. so that these alterations are novel and provided here for the first time). Further, you seem to provide findings from an experimental work, than apparently repeat data that have already been published. The former is great, but I question the appropriateness of simply reproducing data that you imply have already been published. Lastly, the manuscript seems to proceed to a sort of brief review of the literature.

Thus, I think your manuscript needs to be re-structured before peer-review, so that the aims are clear and the overall contents have a logical flow.

Please keep in mind that if you choose to revise and re-submit your manuscript, it will have to undergo the standard peer-review process.

---

## Round 0.2 · Major Revisions

This version of the manuscript is a considerable improvement on the initial submission, and it now provides a more coherent description of the work carried out. Although both reviewers have praised the manuscript, they have raised a number of issues that need to be addressed by a major revision.

Amongst the comments put forward, I agree that the text still needs to be edited to improve its structure as suggested by the reviewers. The Methods need to better identify the different experiments performed. In particular, the subheading "Recent studies" is confusing for the reader, as it suggests that the authors will be referring to previous reports rather than to the provision of new data. Further, as pointed out by one of the reviewers, the methods employed need to be better described.

The text requires careful editorial revision to address typos and grammatical errors. Also, the use of colloquial language is inappropriate and these should ideally be removed (e.g. lines 57-58).

Both reviewers have commented on the issues with the supplementary data, and I agree that the files need to be tidied up so that the data can be adequately interpreted.

Overall, the reviewers' comments are fair, and that by addressing the issues raised the manuscript would be substantially improved.

·

Basic reporting

This manuscript “The Furvela tent-trap Mk 1.1 for the collection of outdoor biting mosquitoes” by Charlwood et al. describes the refinement of the Furvela trap for the collection of anthropophilic mosquitoes. To analyze the new configuration of the Furvela trap for endophagic biters, they compared twice-a-week usage of the tent to daily window-trap collections in Northern Tanzania, and for exophagic biters to outdoor light traps in the PAMVERC trial area in Mozambique. Overall, I believe the trap design to be a good one, especially due to the flexibility of the general design, cost, and due to the ubiquitous nature of CDC light traps. I do think the manuscript suffers from not including more from the large PAMVERC trial data as this has hundreds or thousands of trap nights, and from ambiguity in the methodology.

I fully agree with the premise that human landing catch is becoming unacceptable for use, and its use needs to be phased out. With this, I do feel that a more direct comparison between the Furvela design and HLC should be done. I do not believe this is necessary for publication, but I do think it is necessary to a widespread adoption of the technique.

I do recommend the paper for publication, though think a major revision of the text should be undertaken to improve clarity for the reader. I also feel the format and quality of the figures needs to be improved prior to publication.

Basic Reporting:
The Furvela trap, while previously described, needs to have a basic description in the text to orient the reader to the design. This is especially pertinent as I can’t locate the (what I presume) is the original description of the Furvela trap, as this reference (Charlwood JD: A new efficient and cheap trap for the collection of outdoor-biting mosquitoes. Acta Trop 2005, 95:S1–S506) seems to be to the entire supplementary info covering conference proceedings from the 4th MIM Pan-African Malaria Conference. The supplementary video and SOP do help this, but still I feel the paper would benefit from something brief in the text before the “improvements” part of “The Furvela tent-trap Mk 1.1” section.


Figures as included have the below issues:
• They may not be of sufficient resolution.
• Figure 1 Legend: I would highly recommend listing in the legend what each picture is showing. Defaulting to the text makes description unnecessarily taxing.
• Figure 2 Legend: “Tanzania” is misspelt.
• Figure 2: I don't see an arrow.
• Figure 2: Why is only the blue line labeled?
• Figure 2: The gaps in sampling would be better presented with a broken axis as you then lose information from the time periods you have sampling (i.e. reduce the “white space” in the figure).
• Figures 3 and 4: I'd highly recommend the use of bar graphs over pie charts. It would convey the data in a smaller and more informative package. Additionally, this allows for easier comparison between groups. Stacked barplots may make comparisons between before/after easier. See below figure (in attached pdf) for example (I will also attach the R code to generate this figure). This presentation also allows you to more quickly look between species without having to go back to the figure legend to see what is what.

(see attached PDF for figure)


• Figures 3-4: Make sure that colors are consistent between graphs, i.e. Furvela is always blue.
• Figure 4 Legend: A numbered reference is cited, but the references aren’t numbered.
• Table 2 uses “DRR” which is not defined. The text describes “IRR.”

Other comments:
• Lines 206-207, Where is this time data? If in the supplemental I only see times "1, 2, 3". Was the night divided for collection period? This needs to be said explicitly how the night was split if so.
• Overall, the supplemental data could be cleaned up significantly. i.e. in the indoor_outdoor_furvela document, there are extraneous datasheets that aren’t included in the paper, sheets are labelled poorly (“Sheet 1”, “Sheet 2”, “Collections”, etc..), and some of them have no/poor labeling (no dates, shorthand for species). Fix sheet labels, remove datasets not utilized, and remove figures not used.
• The supplementary file names need to be explicit and the same as described in text, i.e. include “Supplementary File 1 – xxxxxxxx.xlsx”.

Experimental design

The design is significantly muddled by the cobbling together of various studies using the Furvela trap, of which it is poorly described which tents are used (models, etc.), which “improvements” are included, and sampling locations/times/methods compared, etc. I think the authors would benefit from a descriptive table of what sampling was performed.

The methodology needs to be expanded, i.e. number of trap nights, be explicit about what data is re-used from studies published elsewhere, design of the study (i.e. are collectors being rotated through tents, etc.?).

Validity of the findings

A central premise regarding the furvela trap’s uniqueness is that an “entry” behavior is not required for collection. However, as mosquitoes are caught while attempting to enter the hole in the tent (while following the odor plume)---I feel this distinction is subtle from other “entrance” traps. The design is like that of the “infoscitex tent” design we described in Krajacich et al. as mosquitoes collected to odor plumes pushed out of mesh windows are instead collected by suction ports above the mesh (albeit in a simpler design with the Furvela trap).

Mosquitoes have been known to be very adept flyers, entering even small holes in a bed net. While the fan I can imagine makes entry difficult, as there is a large gap in the zipper- how often have collectors found mosquitoes in the tent? How often have you caught full or partially bloodfed mosquitoes in the furvela trap (indicating they were possibly caught while exiting after feeding), and does this proportion differ from that caught in the CDC Light traps?

Additional comments

Line 147: Need a comma after “light-traps”
Typo in line 151, “In Kagera Province, northern Tanzania.” should not be a separate sentence.
Line 214: Remove comma after "In studies"
Lines 224-226: Is it true that only An. funestus and Culex spp. have changed indoor/outdoor ratios? It seems only Mansonia africanus is the same before/after spraying. The two mentioned are the only ones who went from dominant indoor to dominant outdoor, but most changed in proportion. The discussion should address whether any of the vectors (especially those such as Coquelletida fuscopenata that are less-commonly described) have been reported to be more commonly exophagic in other studies (from indoor/outdoor light traps or ideally HLC).

Line 249, need comma between "outdoors" and "suitable"

Line 254: "Window-traps may be a substitute for light-traps." What is this based on? They may be for indoor collections, but obviously not for outdoor collection.

Lines 279-280: Has this light repellence been seen with CDC light-traps with/without the bulb (but having a CO2 source) in this area? This finding is interesting as I am unaware of a documented repellence effect due to light for anthropophilic vectors, though I am aware of many cases of lights attracting vectors (i.e. sandflies) and other non-target dipteran (making sorting more difficult).

·

Basic reporting

The manuscript "The Furvela tent-trap Mk 1.1 for the collection of outdoor biting mosquitoes (#16725)" describes a modified version of the Furvela tent-trap and compare it to other trapping methods.

The English language requires significantly improvement, e.g. have to be checked by a native speaker. I have indicated several sentences which stay unclear for me due to language and grammar issues.

Although the general structure of the manuscript conforms to PeerJ standards, the structure of the introduction is not very easy to follow. Please start with the general background (malaria, number of cases, distribution, ...), then explain the Motivation of your study (indoor collections to not necessarily reflect the outdoor situation, ...) followed by the State-of-the-art (pros and cons of methods used in the past (HLC) and current methods (light traps, tent traps)). Then highlight the lack of knowledge (there are already tent traps available, explain why we need your improved system (entry behavior, ...)) and finally give a description of the explicit study aims (in short: What exactly are you doing and why). In addition, several information are not well referenced.

The figures and tables are generally okay. However, I recommend using bar plots instead of pie charts, which are generally expected to be more comparable between different groups. Some information are missing in the labels of the figure and tables, e.g. explain all abbreviations in table 2 in the label (DDR, TT, CI, adjusted, ...).

Finally, although the raw data are supplied, the data are not very useful. Every table has a different, mostly unstructured format and use different abbreviations, which are not explained.

Experimental design

The research is in the scope of PeerJ, but the explicit knowledge gap and research question is not well described.

As highlighted for the introduction, my critique of missing structure also applies to the methods section. You are comparing the Furvela tent-trap in different settings, but due to the unstructured presentation, it does not become clear what you did in which site. Please give different subsections, e.g. "In the first experiment, we did ..." & "The second experiment was conducted....", etc. In addition, from the description in the methods, it does not completely becomes clear, that the monitoring was conducted after mosquito control interventions. Furthermore, important information on the mosquito sampling (e.g. window traps) and statistical methods (e.g. why negative binomial) are missing.

Validity of the findings

The findings are valid and supported by sufficient data, but are not presented very well. As in the introduction, there is a lack of references for some of the statements and therefore sometimes look like speculation. The research question was not clearly indicated and therefore there is no link between the introduction and the conclusions.

Additional comments

INTRODUCTION
#before line 39: please provide general information on the impact of malaria on public health in Africa (number of cases, distribution, etc.)
#lines 40-42: "This success ..." --> unclear sentence, please rephrase.
#lines 42-44: "Outdoor transmission [...]" --> unclear sentence, please explicitly explain why both, indoor AND outdoor monitoring, is needed, i.e., because indoor monitoring does not necessarily reflect the outdoor situation
#lines 44-50: "Indeed, ..." --> I do not understand why you include this citation. Delete?
#line 52: "untested," --> unnecessary comma
#line 55: "require" --> do you mean "lead to"?
#lines 56-58: "Although ..." --> Please be more specific: malaria vectors = Anopheles spp. or specific Anopheles species (Anopheles gambiae complex). In addition, the vectors for malaria can also transmit other pathogens.
#lines 59-61: "Outdoor ..." --> Please give references for the statement that human landing collections is the best method.
#lines 61-62: "HLC ..." --> Please give a reference.
#lines 62-63: "Such ..." --> Please give a reference.
#lines 63-67: "The objective ..." --> Unclear. As far as I understood, here you want to explain why alternative methods for HLC are needed.
#line 70: What do you mean by "risk-free"?
#line 72: What do you mean by "at least as far as malaria vectors go"?
#line 74: At the first report of a species in the manuscript, please give the full name of the species, e.g. "Anopheles gambiae s.l." instead of "An. gambiae s.l."
#line 75: replace dot by comma: "CDC light-traps." --> "CDC light-traps,"
#lines 93-94: "The ‘Furvela’ tent-trap, ..." --> Why does it not require an entry behavior? Is there a reference of the first description of this trap?
#lines 94-95: "The trap remains ..." --> reference?
#lines 110-112: "We supply ..." --> Was the aim of the study to show that the Furvela trap catach mosquitoes? I thought this was clear before. What exactly are the aims of your study?

METHODS
#before line 116: please give some more information as introduction of the method section, e.g. "in order to improve the Furvela tent-trap Mk 1.1 ..." or information why the improvements were necessary
#line 116: please give a reference for the first description of the Furvela tent-trap Mk 1.1
#line 147: include comma after "In addition to CDC light-traps"
#lines 147- : I completely missing details on the exit window-traps (e.g. construction, references, etc.).
#line 154: "The best ..." --> What trap do you mean? Window-traps or Furvela traps?
#lines 155-156: "As mentioned ..." --> As mentioned before? You have to explain, which risk a linked to HLC and give a reference for the statement.
#line 158: What is "PAMVERC trial"? Explain.
#lines 165-167: Why did you use negative binomial regression, e.g. Why did you not use ANOVA or Poisson regression or .... Please explain how you observed overdisperion in the data, which is one important information to select negative binomial regression as appropriate method. Which explanatory variables were used? Was this statistical method also used to analyze the collected data described in the lines 178-183.

RESULTS
#line 203: comma after "Between the 23/10/2014 and 14/3/2015"
#lines 205-206: "r = 0.93, P = 0.0003" --> What kind of statistical method was applied here? Spearman? There is no explanation here or in the method section.
#lines 206-208: "A larger proportion ..." --> As far as I understood from the method section, you only aimed to compare the different kind of traps? Why did you also analyze the temporal pattern? Explain in the description of the study aims in the introduction and explain in the methods how you analyzed the temporal pattern.
# lines 208-212: "Immediately following the application ..." --> Please explain this mosquito control measurements in the method section and not in the results section.
# lines 214-216: "In studies, from Tanzania, ..." --> Please move to the discussion section.
# lines 216-217: "The other species ..." --> Which other species? Describe.
# lines 217-219: "Comparable results were ..." --> What do you mean by "comparable results"? Describe.
# lines 233-234: "The incident rate ratio ..." --> I do not understand what you are doing here. Give more information on the IRR and the variables TT and L.
# lines 234-236: "Variables including collector, ..." --> The modelling approach must be explained in the method section.
# lines 234-236: "Variables including collector, ..." --> What do you mean by the variable "collector". Might be better included as a random effect?
# line 239: "Anopheles" in italics

DISCUSSION
#general: Please also discuss potential problems with the tent traps. For example, please discuss the problems of the sampling standardization, which might be highly dependent on the collector in the tent.
# lines 247-248: "Because of the risks ..." --> Give a reference.
# lines 258-249: "In order to determine ..." --> You do not determine the transmission, but the density of potential vectors. Please rephrase.
# lines 249-252: "The ratios ..." --> In general, the comparison does not give any information about the suitability of the Furvela tent-trap to trap outdoor biting mosquitoes, e.g. because you do not have information about the size of the real outdoor biting mosquito population, but indicate a similar performance like the light-trap. Please rephrase.
line 254: "Window-traps may be a substitute for light-traps." --> What do you mean by "substitute"? I do not understand.
#line 258: "sp." --> not in italics. In addition, please use "spp." as there are several species present in Cameroon.
#lines 267-268: What do you mean by "chaotic"? Spatial-temporal variability? Please explain and rephrase.
#line 269: "indoor light-trap" instead of "light-trap"
#line 269: "For example" instead of "Thus"
# line 277: "With the current ..." --> You mean the elimination of mosquitoes in order to eliminate malaria. Rephrase.
# lines 285-288: "Ease of transportation ..." --> Why are vector maps needed? Explain.
# lines 288-289: "Such information ..." --> How do the tent traps help to estimate the flight range of mosquitoes? Completely unclear for me. Explain.
# lines 289-290: "They may ..." --> How to tent traps help in the regarded to determine the intervention area? Explain.
# lines 307-308: "Presently this ..." --> Why is there a restriction to indoor insects? Explain.
# line 308: "The use ..." --> I do not understand the sentence. Rephrase.

---

## Round 0.3 · Minor Revisions

Further to the reviewer's comments:

* Lines 69-71. It is very difficult to make sense of this sentence.

* Line 178. It should read "at four-hour intervals" rather than "four hourly intervals"

* Lines 178, 179, 206, etc – The time formatting needs to be consistent across the manuscript. Similarly, the formatting of dates also needs to be consistent.

* Lines 207-208 – It would be helpful for the unfamiliar reader if IRR and DRR were either explained or the appropriate references provided (or both).

* Methods – It is important to state who was responsible for the taxonomic identification of mosquitoes, and preferably also describe the tools used and where this was done.

* Line 229 – "was registered"

* Line 235 – Please clarify if consent was verbal or written (or both)

* Line 242 – Best to provide the p-value as p<0.001 for consistency.

* Line 246 – Best to provide the p-value with 3 decimal cases for consistency.

* Figure 4 – Two issues here. First, do the letters on the x axis refer to the same species as those listed in Figure 3? Either way, it is important that the species are also listed in the legend for Figure 4, so that it can be understood on its own. Secondly, it is confusing that you have A & B in the x axis but also use A & B to identify the two panels. Probably best to refer to the panels as 4A & 4B, and state that in the figure legend, i.e. "Panel 4A: collections…"

* Table 1 is not self-explanatory, and needs some revision:
- It is best to make it clearer and say "number of light-trap collections".
- What does the column heading "N" mean? Is it number of light-trap collections as well? If so, please state it as it is confusing.
- The table legend needs to describe what the data actually are. I assume you refer to number of specimens per collection, and this should be clearly explained.
- There is inconsistency in the number of decimals provided for the mean and ratios and their respective CIs. Probably best to use two decimal places throughout, except for p-values for which 3 decimal places are ok.
- "spp." and "s.l." should not be italicized. This also applies to other figure legends and the manuscript in general.
- by "density ratio" do you mean DRR as in Table 2? Please make the terminology consistent across the manuscript.

* Table 2:
- For consistency with Table 1, please provide all values with 2 decimal cases, but p-values with 3 decimal cases.
- I don't understand the 95% CIs here. What do they refer to? In each column you have two values provided in two rows, but just one CI.

* I concur with the reviewers that the Discussion can be difficult to follow and is somewhat disjoint. In the very least, the first paragraph of the Discussion is way too long and would benefit from having a couple of paragraph breaks inserted to make it easier for the reader.

* Line 318 – Best to say something like "improving capture efficiency" rather than "enhancing collections" which is unclear.

* Line 321 – Please delete "(including JDC’s old socks)", which is funny but not really appropriate or scientifically informative.

* Lines 367-8 – This sentence ("These mosquitoes…") has a statement of fact that should be backed up by the appropriate reference(s).

* Line 380 – Please spell out "ms".

* References: There is considerable inconsistency in the formatting of the references. These need to be made consistent and in accordance with the journal's guidelines.

·

Basic reporting

I feel the manuscript has been greatly improved since the initial version, both in terms of clarity and the overall formatting. I feel that the changes that have been incorporated are appropriate and sufficient.

Below I list some minor corrections/suggestions.
Lines 36-39: This sentence is a bit awkward/run-on and would be better split into two. Also merge into below paragraph.
Line 239: Report week numbers or change figure to show dates
Line 242: Report density drop
Line 278: Remove “really”
Line 319-321: I’d mention explicitly that light being a repellent has not been shown previously.

Overall comment: The discussion is somewhat disjointed and doesn’t flow easily from paragraph to paragraph. I’d suggest some improvement of transition sentences between paragraphs, and the paragraph order. I think it’s publishable in this format, but could be improved.

Experimental design

no additional comment

Validity of the findings

no additional comment

·

Basic reporting

no comment

Experimental design

no comment

Validity of the findings

no comment

Additional comments

The authors significantly improved the manuscript and can be accepted. However, to my opinion, the supplementary tables still need significant improvement. Especially please do not use unexplained abbrevations (e.g. "coll" = collection method, "afr" = "Anopheles funestus", AG part = ???, ODDS = ???). Please tidy these tables up, there are a lot of blank cells or unused columns.

---

## Round 0.4 · accepted · Accept

Please note however, that my previous comments regarding Table 2 have not yet been addressed. As a result, the manuscript has been accepted on the condition that the issues raised below will be addressed prior to publication:

- For consistency with Table 1, please provide all values with 2 decimal cases, but p-values with 3 decimal cases.
- I don't understand the 95% CIs here. What do they refer to? In each column you have two values provided in two rows, but just one CI.